# Influence of building collapse on pluvial and fluvial flood inundation of metro stations in central Shanghai

Zhi Li[1], Hanqi Li[1], Zhibo Zhang[1], Chaomeng Dai[1], and Simin Jiang[1]

[1]College of Civil Engineering, Tongji University, Shanghai, China

**Correspondence:** Zhi Li (zli90@tongji.edu.cn)

**Abstract.** Urban flooding poses a significant threat to vulnerable underground infrastructure systems, such as metro stations. Building collapses induced by earthquakes alters urban building layout and coverage, consequently influencing flood inundation and propagation patterns. This study employs GPU-accelerated hydrodynamic simulation to investigate the mechanisms by which building collapse affects subsequent pluvial or fluvial flooding in the Huangpu district of Shanghai. Massive building collapse layouts are randomly generated, on which hydrodynamic simulations are performed and the inundation process of the metro stations are analyzed. The results reveal that pluvial floods are strongly influenced by localized topography distributed across the city. Consequently, building collapse has a more substantial impact on pluvial flooding when more buildings are collapsed. In contrast, fluvial floods are sensitive to the source location (e.g., location of levee breach) and the long travel route. Building collapse can either positively or negatively influence fluvial flooding by constricting or blocking the flow path. This work highlights the complex mechanism of earthquake-flood multi-hazard processes, emphasizing the importance of performing local-to-local analysis when both the hazard (e.g., individual building collapse, fluvial flood) and the hazard-bearing body (e.g., metro station) are localized. To better serve urban disaster prevention and mitigation, more efforts should be directed on developing physics-based high-resolution urban earthquake-flood simulation methods, as well as on acquiring data to drive such simulations.

## 1 Introduction

Climate change and urbanization have increased human exposure to urban flood hazards, particularly in the East Asia (Cao et al., 2022; Rentschler et al., 2023). To enhance urban resilience to flood hazards, there is an urgent need to understand the physical processes of flood propagation under various current and future scenarios. Hydrodynamic modeling is an important tool for urban flood studies. Recent advancements in high-performance parallel computing (HPC) have facilitated city-scale, high-resolution (on $< 5m$ resolution digital elevation model, DEM) rapid hydrodynamic simulations (Guo et al., 2021; Morales-Hernández et al., 2021; Sanders and Schubert, 2019; Schubert et al., 2022a). Combining HPC technology and traditional hydrodynamic models, the high resolution spatial-temporal evolution of the inundation depth and extent can be reproduced or predicted, thereby aiding in the design and improvement of flood mitigation strategies.

When considering urban flood resilience, metro systems are a weak link. Underground infrastructures are inherently more vulnerable to floods, and evacuating passengers from inundated metro tunnels poses immense challenges. Megacities typically

have well-developed metro systems, and flooding of the urban metro system has been reported worldwide (Aoki et al., 2016; Forero-Ortiz et al., 2020; Toda et al., 2009). In China, megacities including Beijing, Shanghai, Guangzhou, Wuhan and Shenzhen have more or less experienced flooding of their metro lines (Lyu et al., 2019, 2020; Wang et al., 2021). In July 2021, a tunnel of the Zhengzhou (China) metro Line 5 was inundated due to heavy rainfall and inadequate emergency response measures, resulting in 14 fatalities (Yang et al., 2022). This devastating incident has raised nationwide concerns regarding comprehensive flood risk evaluation and the urgent need for robust mitigation strategies tailored specifically for urban metro systems.

Modern metro stations and tunnels feature multiple flood prevention equipment such as flood gates, sand bags and pumps. The entrance of the metro stations are often elevated to avoid flooding. These flood prevention measures are expected to work well under most circumstances. However, it remains unclear if these measures are fully functional under extreme multi-hazard scenarios. Multi-hazard refers to scenarios where a system or infrastructure (e.g., a metro station) is subjected to multiple hazardous events, either concurrently or sequentially. The compounding effects of multi-hazards can potentially lead to more severe consequences compared to the impact of individual hazards. Compound flood is a common multi-hazard scenario. In 2013, typhoon Fitow triggered heavy rainstorm in Shanghai, China. At the same time, fluvial flood from the upstream attacked Shanghai. An astronomical high tide further exacerbated the compound flood, which finally caused 2 deaths and a loss of approximately 150 million US dollars. Geological hazards also compound the impacts of urban floods (Gill and Malamud, 2014). For example, it has been shown that land subsidence exacerbates flood depth and extent (Johnston et al., 2021; Navarro-Hernandez et al., 2023). Another example is the sequential occurrence of a magnitude 7.8 earthquake in April 2015 and a severe flood event in August 2017 within the same region of central Nepal. The infrastructure damaged during the seismic event had not been fully rehabilitated when the flooding occurred, leading to more devastating damage compared to if the area was attacked by a single disaster (Gautam and Dong, 2018). For coastal cities, earthquake could induce tsunami, which turns into coastal floods. After the earthquake, the flood resilience could be compromised due to two main factors: (i) the flood-prevention structures (e.g., flood walls) and equipment (e.g., pumps) could be damaged during the earthquake, and (ii) the collapsed buildings could block the evacuation routes. The confluence of these two factors can exacerbate the severity of coastal flooding impacts (Goda et al., 2019; Ito et al., 2020; Román-de La Sancha et al., 2022; Takabatake et al., 2022).

Urban multi-hazard studies have attracted increasing attentions in recent years. However, existing studies often focus on risk assessment that is based on historical data or simplified simulations (e.g., Depietri et al., 2018; Owolabi and Sajjad, 2023). With the advances in HPC and urban flood simulation, it is possible to reproduce large-scale, high-resolution flood propagation process under multi-hazard scenarios. This allows for an investigation on the physical mechanisms how multi-hazards affect urban flooding compared with pluvial or fluvial flood hazard alone. With better depiction of the multi-hazard flooding process, risk assessment methodologies can be enhanced too.

In this work, we use high-resolution hydrodynamic simulations to estimate the evolution of pluvial/fluvial flood depth at 17 metro stations in the Huangpu district located in the central region of Shanghai. We compare the flooding dynamics in the actual Huangpu district to scenarios that incorporate multiple random realizations of building collapse. In these simulated scenarios, certain buildings are assumed to have collapsed, obstructing roads and altering the urban layout. However, we do

not directly model the building collapse processes. Instead, building collapse is represented as the spreading of the debris that changes the building height and spatial occupancy. The total volume of a building remains unchanged before and after it collapses. For a more detailed description of how building collapse is treated, please refer to Section 2.3 and Takabatake et al. (2022). This approach serves as an approximation of post-earthquake conditions, enabling the analysis of the multi-hazard process involving the cascading impacts of seismic activity and subsequent inundation. Prevailing urban flood studies focus on the global inundation depth and extent. However, collapsed buildings and flooded metro stations are localized. It is rarely investigated that how do localized building collapse affect the global flood propagation, and then subsequently affect local inundation of metro stations. In this study, we aim at understanding if, how and why building collapse affects the inundation of the metro stations, with a focus on the spatial heterogeneity of both the building collapse and the station water level, thus addressing the local-to-local multi-hazard process. The findings will serve as the cornerstone for multi-hazard risk assessment of urban metro systems. However, it should be noted that (i) our analysis focuses on the hazard rather than the risk aspect of building collapse and urban flood. An in-depth exploration of improved risk assessment techniques is not discussed herein, and (ii) building collapse is used to represent the consequence of earthquake, but the physical processes of seismic events are not explicitly modeled. With this aim, topics such as how the earthquake is triggered, how the buildings are collapsed, how the flood enters the metro station and how to assess the multi-hazard probability and risk are beyond the scope of this manuscript.

The rest of this manuscript is arranged as follows: Section 2 describes the study site (Section 2.1), the numerical simulation methods (Section 2.2) and the building collapse scenarios (Section 2.3). Section 3 reports and analyzes the simulation results. Section 4 discusses the implications, limitations and future works of this study. Section 5 draws the conclusions.

## 2 Methods

### 2.1 Study Area

The study area is the Huangpu district located in the center of Shanghai. Shanghai is one of the largest cities in China, with a population of approximately 25 million. Huangpu district has an area of about 20 square kilometers and a population of 0.66 million. Huangpu district is bounded by the Suzhou River in the north and the Huangpu River in the east and south. The west boundary connects the Jing'an and Xuhui districts of Shanghai (Fig. 1). As of 2023, the Shanghai metro system has 19 metro lines, in which 8 lines pass through the Huangpu district, forming 17 stations (labeled as station 1 to 17 in Fig. 1).

Multiple drivers could lead to or aggravate flooding in Shanghai, including typhoon, astronomical high tide, upstream fluvial flood, low-lying topography, sea level rise and land subsidence. These factors make Shanghai one of the most vulnerable cities to flood in the world (Balica et al., 2012). Shanghai adopts four levels of flood defense systems: the sea walls, the river flood walls, the pump stations and the pipe drainage system. The current defense system is generally effective, but might be inadequate under occasional extreme conditions or future climate change scenarios (Ke et al., 2021; Yin et al., 2021; Zhou et al., 2017). For example, Shanghai experienced a short but intense rainfall event on July 21th of 2023, with a maximum rainfall rate exceeding 100mm per hour (Fig. 3), leading to mild waterlog. The reported inundation depth reach 25cm at multiple locations

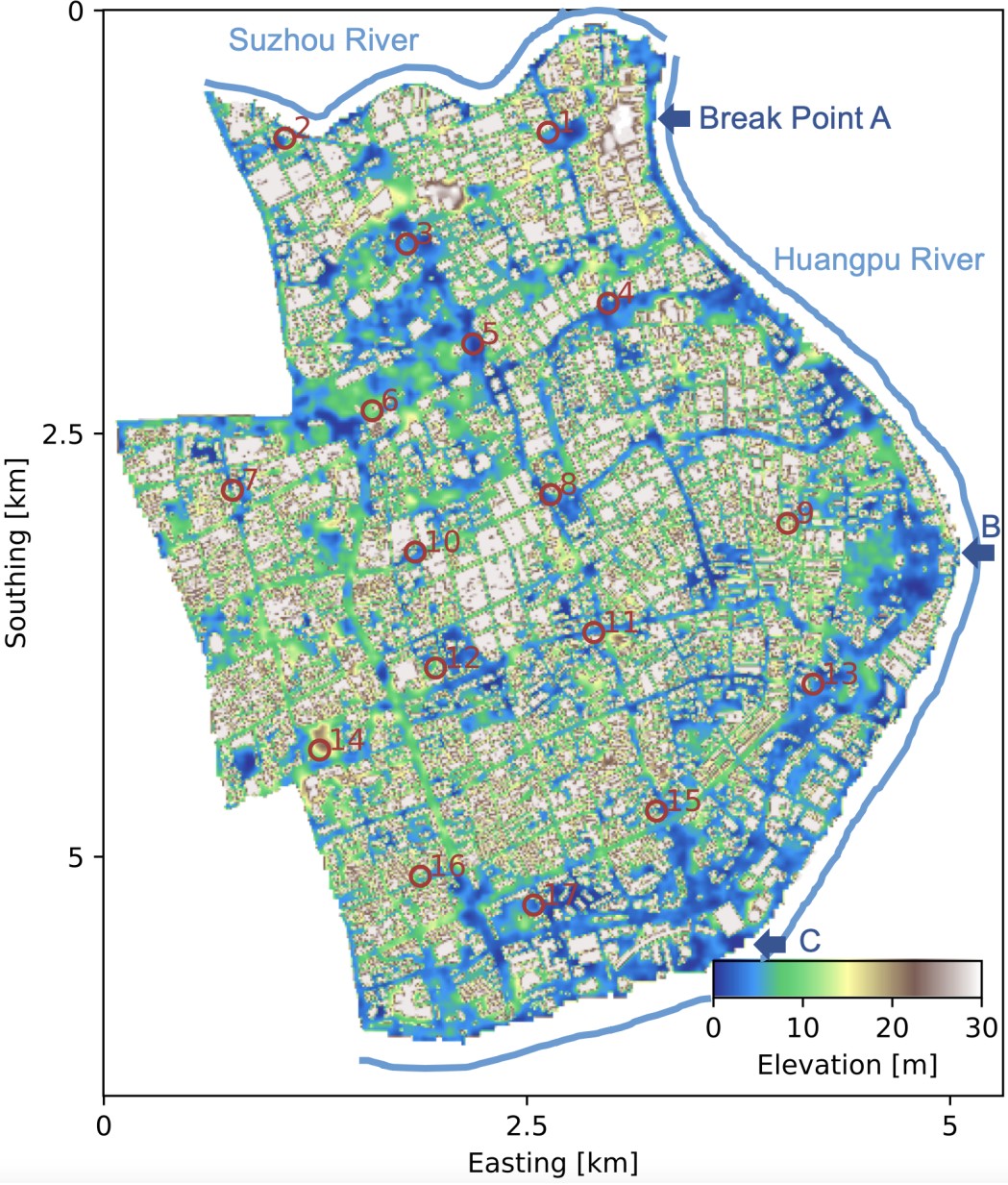

**Figure 1.** The DEM of the study area. The red markers indicate the 17 metro stations in the study area. The dark blue arrows are the hypothetical break points of the flood wall. For better visualizations, buildings are assigned with a uniform elevation of 29m on this figure, but their original elevation is used in the simulations.

in the city. It remains important to better understand the flooding mechanisms in Shanghai and keep improving flood prevention system and flood response strategies correspondingly.

## 2.2   Numerical Simulation

In this work, the flooding of the Huangpu district is simulated with the SERGHEI-SWE model (Caviedes-Voullième et al., 2023). SERGHEI-SWE is an open-source, high-performance hydrodynamic model. It solves the two dimensional shallow water equations (SWE) on Cartesian grids using Godunov-type finite-volume methods (Morales-Hernández et al., 2021). SERGHEI-SWE is a parallel SWE solver that achieves performance portability across a variety of computational backends. Without modifying the source codes, SERGHEI-SWE can perform parallel computation on a CPU through OpenMP, or on a Graphical Processing Unit (GPU) through either CUDA (a parallel computing model for Nvidia GPUs) or HIP (a C++ kernel language for parallel computing on Nvidia and AMD GPUs). This feature is achieved through the Kokkos framework (Trott et al., 2022). It also supports distributed memory parallelization through the Message Passing Interface (MPI), which allows parallel computation across multiple CPU or GPU nodes. GPU computing has dramatically reduced the computational cost for large-scale hydrodynamic simulations. Caviedes-Voullième et al. (2023) have shown that using SERGHEI-SWE, for a computational domain with 0.5 million grid cells, hydrodynamic simulations on a workstation GPU remain faster than that on 128 CPU threads. Thus, SERGHEI-SWE with GPU computing capability is an ideal tool for completing massive flood simulation scenarios (e.g., the multiple random building collapse scenarios used in this study) within an acceptable time frame.

To build the hydrodynamic model for the Huangpu district, data on the DEM, surface roughness and boundary conditions are required. Flood simulation results are sensitive to the DEM (Xu et al., 2021). The 30m resolution DEM of the study area is freely available online, but this resolution is too coarse to resolve individual buildings and road sections, which have significant influence on the flood propagation pathways. Thus, the DEM is downscaled onto 5m grids, resulting in 1.36 million grid cells. It should be noted that DEM downscaling does not improve the accuracy of the DEM itself, but allows for better delineations of the building outlines. The building shape and height data are integrated into the DEM, replacing the original coarse-resolution building pixels. The final DEM can be visualized in Fig. 1.

The road layout data is integrated into the land use file, which provides spatially-distributed surface roughness for computing the bottom drag. In this work, the land use of the study area is simplified to three types: buildings, roads and others. They are assigned with a Manning's roughness coefficient of $0.012 s/m^{1/3}$, $0.016 s/m^{1/3}$ and $0.03 s/m^{1/3}$ respectively. Arguably, 5m grid resolution does not resolve the road shoulders. Existing approaches such as the artificial porosity model and the elevated edge model could identify the subgrid-scale road shoulders (e.g. Guinot et al., 2018; Hodges, 2015; Li and Hodges, 2019), but they are not adopted herein because the focus of this manuscript is on comparing the pre- and post-collapse scenarios. Indeed, the influence of the road shoulders on flood propagation does not depend on the building collapse status. Building drag force is neglected in the simulations. Rainfall is allowed to fall on the building roof, then flow to its neighboring cells following the topographical gradient. Detailed roof storage or drainage devices are not considered due to the lack of data. Soil infiltration and drainage through the stormwater pipelines are both ignored. Admittedly, this would overestimate the flood inundation depth. However, under extreme flood events, the soil and the drainage pipes are likely saturated, meaning that they have minor influence on surface hydrodynamics (e.g., Wilfong et al., 2024). More detailed discussions on the roles of infiltration and drainage can be found in Section 4. The computational domain is initially dry. The west boundary of the domain is open and

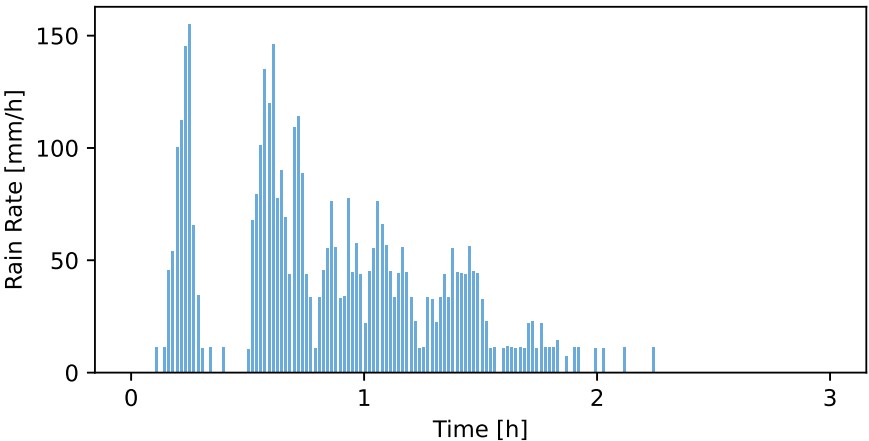

**Figure 2.** The rainfall data recorded at the Tongji University main campus between 2:30pm and 5:30pm, July 21th of 2023.

water is allowed to exit the domain freely. All other boundaries are closed to represent fully functioned flood walls along the rivers.

The flood simulations conducted in this study encompass two distinct scenarios: fluvial and pluvial flooding events. To model fluvial flood, assume three locations on the flood wall are breached (e.g., due to earthquake) and a constant water depth of 3m is enforced at the three breakpoints (marked on Fig. 1). To model pluvial flood, assume the rainfall rate equals the recorded precipitation on July 21th of 2023 (Fig. 2). Both scenarios are modeled for 3 hours. The inundation extent, depth, particularly the depth at the 17 metro stations, will be monitored and analyzed. It should be noted that a real metro station often has multiple exits, each with different elevation and orientation that cannot be adequately resolved at the grid resolution we use. Thus, in our study, we do not treat metro stations as sink terms or outflow boundaries. We only observe the water depth of the grid cell where the metro station is located. The possibility of further refining the treatment of metro stations will be discussed in Section 4.

## 2.3 Building Collapse Model

The earthquake risk of Shanghai is generally low. Historically, Shanghai rarely experiences an earthquake with a surface-wave magnitude greater than 7.0. However, the large population density, tunnel constructions in the soft soil, massive old civil houses as well as more than 150 skyscrapers (i.e., buildings taller than 150m) make Shanghai vulnerable to potential earthquake attacks. It is estimated that in the Nanjing Road (located in Huangpu district) where plenty of old houses are made of brick and timber, 7.2% of buildings will be seriously damaged under an intensity VI earthquake (Chinese Intensity Scale), and 24.8% of buildings will be seriously damaged under an intensity VII earthquake (Cole et al., 2008). Given the scarcity of seismic data in Shanghai, the lack of detailed building information, as well as the lack of robust physical models that simulate the building collapse process, we did not directly quantify building collapse based on seismic magnitudes. Instead, we use a similar approach to Takabatake et al. (2022), where building collapse is modeled by distributing certain amount of

150 the "building material" to its surroundings, thereby increasing the spatial occupancy and reducing the height of the building. In this study, we assume that when a building collapses, (i) its spatial occupancy doubles, and (ii) its total volume remains unchanged. The increased spatial occupancy (i.e., the spreading of the "debris") is distributed uniformly around the building. The reduced height of the building can be estimated using the volume and the new spatial occupancy. A special case exists when two buildings are adjacent and their spatial occupancy overlap when collapsed. In such situations, the two buildings are

155 treated as one single building when calculating the debris extent. It should be noted that if two buildings with different heights are treated as one building, the final building height after collapse could be greater than its original height, which is unrealistic (Fig. 3). However, this phenomenon has negligible influence on the subsequent flood simulation because as long as the building debris is not inundated, slight variation of its height has minor influence on the flow field. That is, the obstruction of the flow path (due to the spreading of the debris extent) has much stronger influence on flood propagation than minor changes of the

160 debris height. With this approach, building collapse is only a simplified representation of the consequence of hypothetical seismic events, but the physical mechanism connecting earthquake and building collapse is not involved. Although it is an idealized representation that neglects factors such as building strength and seismic force magnitude, this approach effectively characterizes the obstruction and alteration of flood propagation paths by collapsed buildings, which is the primary focus of this study. The simplified modeling of post-collapse building geometry captures the essential effects on flood dynamics without

introducing unnecessary complexities.

To compensate for the absence of deterministic earthquake – building collapse model and data, we generate $n$ hypothetical building collapse scenarios with $r\%$ randomly selected buildings collapsed (Fig. 4). From modeling perspective, this is equivalent to perform hydrodynamic simulations with $n$ different DEM and roughness inputs representing various building damage scenarios. Herein, $n = 100$ and $r = 10$, 20 or 40 respectively. By analyzing hydrodynamic simulation results from 100 possible

building collapse scenarios, the inundation patterns of the metro stations can be statistically characterized. Then, typical building collapse and metro station inundation scenarios will be examined in more detail, revealing deterministic flood propagation mechanisms underlying the statistical observations.

## 3 Results

Before illustrating the simulation results, it is necessary to define a few quantitative metrics that aid the analysis. Herein, three

metrics are defined: the average deviation of water depth ($\epsilon_{\text{avg}}$), the percent of flooded scenarios ($p_{\text{flooded}}$), and the percent of the aggravated scenarios ($p_{\text{agg}}$, see Eq. 1-3).

$$\epsilon_{\text{avg}} = \frac{\sum_{i=1}^{n} \left[ \overline{h_i(t) - h_{\text{ref}}(t)} \right]}{n} \tag{1}$$

$$p_{\text{flooded}} = \frac{n_{\text{flooded}}}{n} \tag{2}$$

$$p_{\text{agg}} = \frac{n_{\text{agg}}}{n_{\text{flooded}}} \tag{3}$$

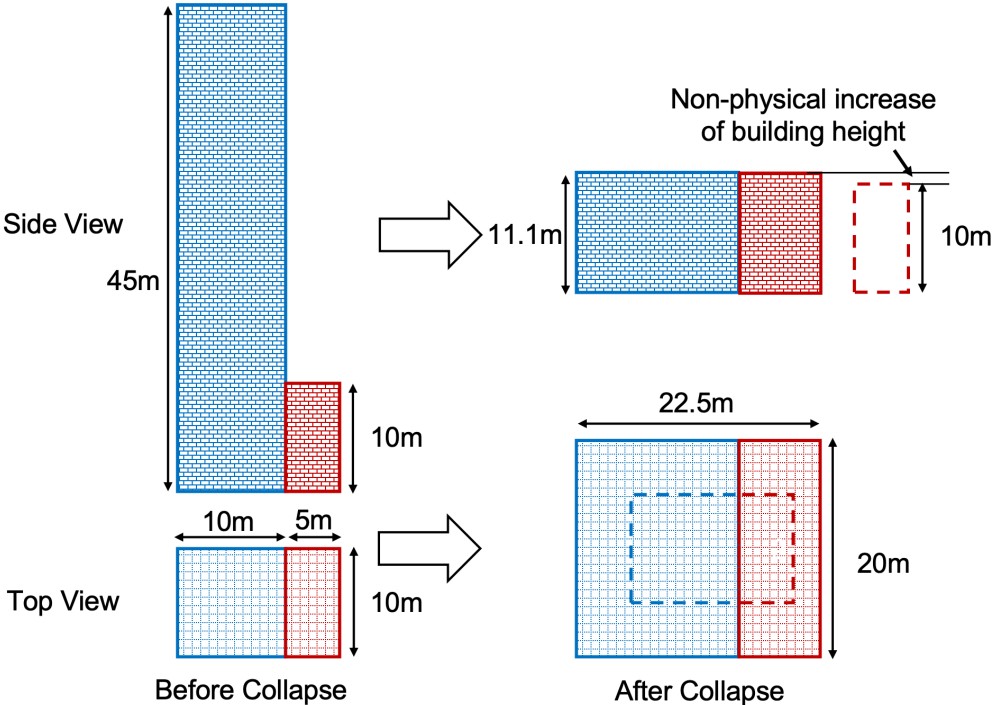

**Figure 3.** An illustration of the building collapse model when two buildings are adjacent. The total volume of the two buildings remain unchanged after collapse. The spatial occupation increases, whereas the height decreases and causes a non-physical increase in the height of the shorter building.

In Eq. (1)-(3), $h_{ref}(t)$ is the simulated depth for the reference scenario (i.e., without building collapse) at a given metro station, $h_i(t)$ is the simulated depth of the $ith$ random building collapse scenario, the overbar indicates time averaging over the 3-hour simulation period, $n = 100$ is the total number of testing simulations, $n_{flooded}$ is the total number of testing scenarios in which a given station is flooded (i.e., the maximum water depth exceeds 1cm) and $n_{agg}$ is the number of testing scenarios with greater flood depth than the reference scenario at the given station.

## 3.1 Pluvial Flood

Figure 5 shows the inundation depth at metro stations 4, 5, 15 and 16 under pluvial flood scenarios. The thick black curve represents the reference scenario without building collapse. The red curves are the 100 random building collapse scenarios. Other stations are either not flooded, or the water is too shallow to cause any catastrophic impact. It can be seen that building collapse has influence on the inundation depth at all four stations displayed. For each station, different building collapse scenarios could either increase or decrease the inundation depth compared with the reference scenario. Station 4, 5 and 15 show similar inundation patterns, where the water depth is small initially, then surges to tens of centimeters. It indicates that these stations are located in low-lying areas that receive water from their surroundings. Station 16 is different in that the water

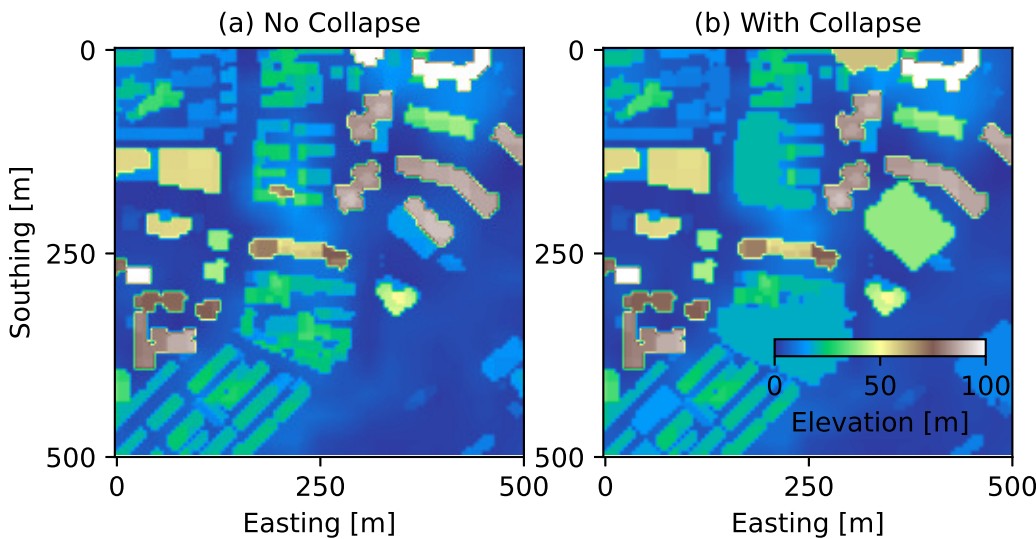

**Figure 4.** A comparison between (a) the original DEM and (b) the DEM with a few buildings collapsed in one of the $n$ random building collapse realizations in the study area.

depth declines when rainfall weakens after about 90 minutes, meaning that station 16 both distributes water to its surroundings, and receives water from its surroundings. The latter is evidenced by the different depth evolution patterns with respect to
different building collapse scenarios.

Interestingly, Fig. 5 shows that, at station 5, despite most building collapse scenarios exhibit an increasing water depth to about 60cm, a few scenarios are almost not flooded (characterized by the near-zero water depth). Detailed examination (not shown) reveals that station 5 is close to a building. When this building is randomly chosen to collapse, station 5 is completely buried and is unaffected by flood. This situation is not impossible in the real world because there exists metro station entrances
inside the buildings in Shanghai. When such buildings collapse to bury the entrance, although flood water is more difficult to invade the station, passengers are more difficult to evacuate too. It implies that for a complete analysis of multi-hazard risk of the metro stations, the locations of each entrance should also be considered separately because the location affects station vulnerability under earthquake-induced building collapse, but vulnerability analysis is beyond the scope of the present study.

As more buildings are collapsed ($r$ increases), the envelopes of the inundation depths expand, implying that more and
more building collapse scenarios diverge from the reference scenario. It can be seen from Table 1 that $\epsilon_{\mathrm{avg}}$ increases with $r$ at all three stations, indicating that the change of building layout (due to building collapse) has strong impact on the flood water propagation. The same conclusion was also mentioned in Bruwier et al. (2020). At all three stations displayed and with all $r$ values tested, 100% of the testing scenarios are flooded. That is, although building collapse affects the magnitude of flooding at metro stations, it does not change the inundation status. This is in contrast to the case of fluvial flood, which will
be discussed later in Section 3.2. The $p_{\mathrm{agg}}$ values exhibit different behaviors as $r$ increases at the three stations. For station 4,

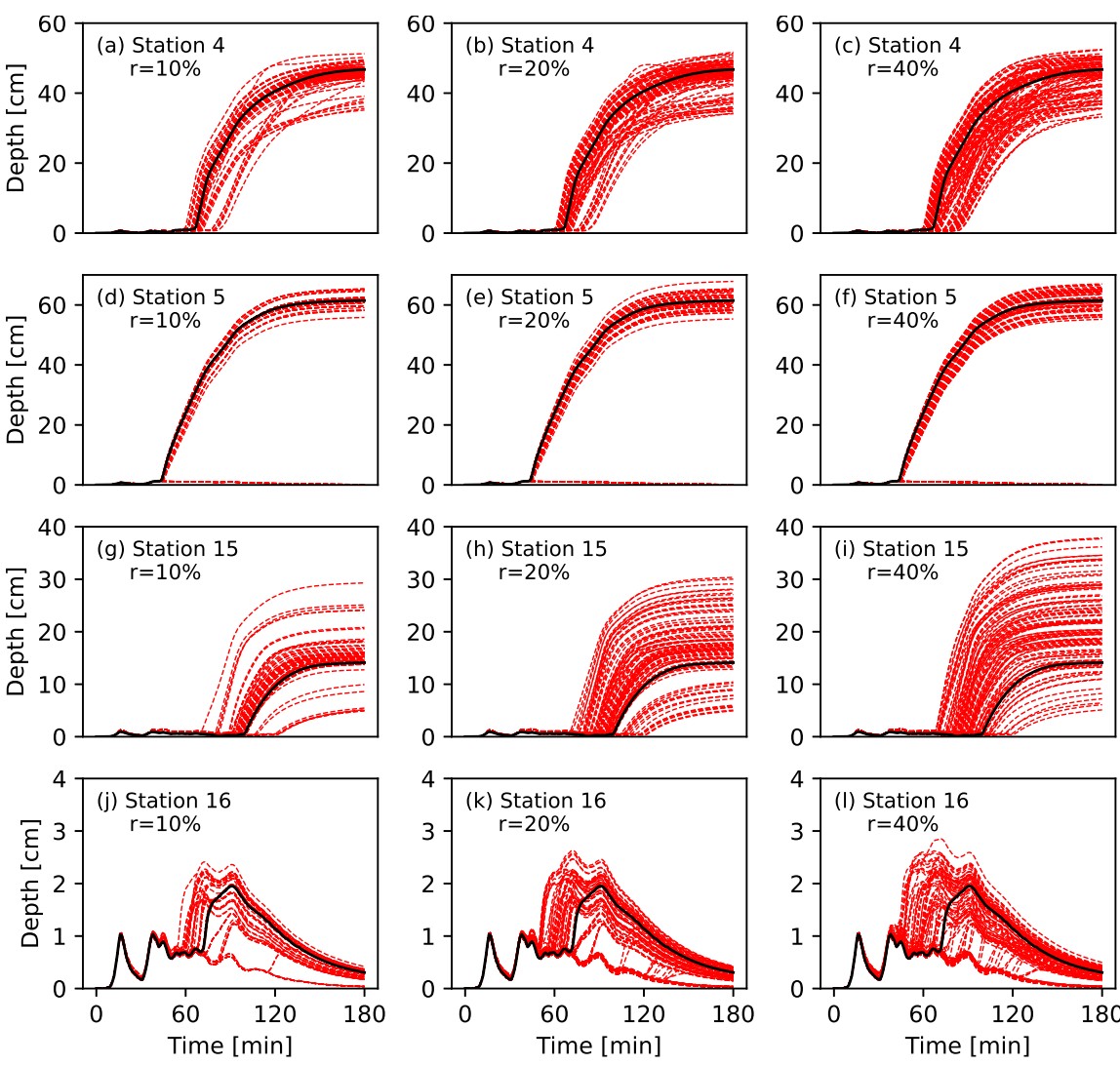

**Figure 5.** Water depth evolution at metro station 4, 5, 15 and 16 for the pluvial flood scenarios. The black line is the original DEM without building collapse. The red lines are the 100 random realizations with buildings collapsed.

building collapse generally alleviate the flood risk as $p_{agg}$ remains less than 50%. Station 15 shows the opposite behavior (with $p_{agg} \geq 80\%$ ), where building collapse mostly enhance flood risk. For station 5, $p_{agg}$ decreases from 67% to 26% as $r$ increases. Clearly, it is difficult to evaluate the increased/decreased flood risk from an ensemble point of view, as it is the specific buildings that collapse (which are different for different scenarios) determine the subsequent flood propagation and water accumulation
patterns.

**Table 1.** The mean deviation from the reference scenario, the proportion of scenarios being flooded, and the proportion of scenarios with aggravated flood disaster (i.e., greater depth than the reference) among the 100 random building collapse and pluvial flood scenarios.

| | Station 4 | | | Station 5 | | | Station 15 | | |
|---|---|---|---|---|---|---|---|---|---|
| | $r =10\%$ | $r =20\%$ | $r =40\%$ | $r =10\%$ | $r =20\%$ | $r =40\%$ | $r =10\%$ | $r =20\%$ | $r =40\%$ |
| $\epsilon_{avg}$ [cm] | 1.43 | 2.54 | 3.42 | 4.16 | 9.35 | 13.6 | 1.22 | 2.99 | 4.95 |
| $p_{flooded}$ [%] | | | | | 100 | | | | |
| $p_{agg}$ [%] | 45 | 32 | 36 | 67 | 37 | 26 | 85 | 80 | 85 |

To further uncover the mechanisms how building collapse affect inundation, three scenarios are selected and analyzed in detail, which are the reference scenario (no building collapse) and the scenarios with the largest and smallest inundation depth (30cm and 5cm respectively) at metro station 15 ($r =20\%$). As can be seen from Fig. 6, with pluvial flood, water accumulates in local topographic depressions, resulting in discrete inundated areas across the study region. In all three scenarios, an inundation
patch is present in the vicinity of station 15, posing a potential flood risk to this metro station. However, Fig. 6(d) sees buildings collapsed on the northwest of station 15, leading to less depression area that stores water (comparing Fig. 6c with Fig. 6a) and reduced road width between buildings that enhances flow speed (comparing Fig. 6d with Fig. 6b). Both factors facilitate flood propagation towards station 15. On the contrary, in Fig. 6(f), building collapse occurs away from station 15, which has minor influence on the inundation of station 15 because pluvial flood is strongly affected by the local topography. Clearly,
the influence of building collapse on pluvial flood propagation is highly localized. Flooding of a metro station is not expected to aggravate if the buildings near the station do not collapse. This finding also explains why $p_{flooded}$ stays at 100% under all circumstances (Table 1), as the localized influence of building collapse can hardly affect the city-wide occurrence of pluvial flood.

## 3.2 Fluvial Flood

Figure 7 shows the inundation depth at metro stations 4, 13 and 17 under fluvial flood scenarios. Other stations are either not flooded, or the depth is too small (e.g., less than 1cm). This is expected because the three stations displayed are close to the break points of the flood wall (Fig. 2). Station 4 is not flooded in the reference scenario, and hardly flooded when $r =10\%$ and 20%. Only a few realizations exhibit non-negligible inundation depth (Fig. 7b). When $r$ is increased to 40%, 17% of the 100 random realizations are flooded (Table 2). Clearly, building collapse enhances the flood exposure of station 4 and the exposure
exacerbates as $r$ increases. Station 13 exhibits the opposite behavior, where a large $r$ reduces flood exposure. At $r =40\%$

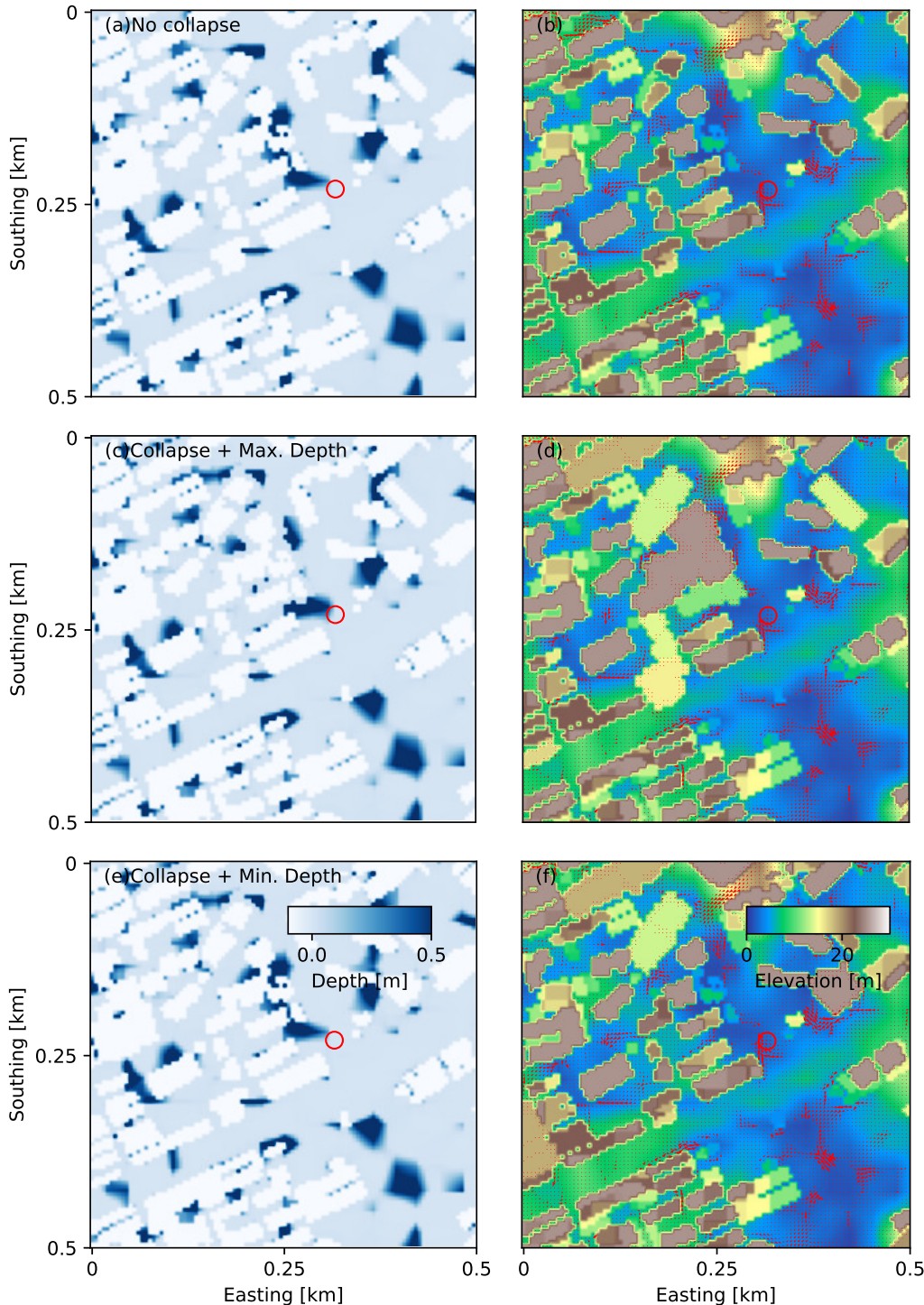

**Figure 6.** Simulated water depth (left column) and velocity (right column, on top of DEM) at 1 hour for selected pluvial flood scenarios near metro station 15 (r=20%). The top row is the reference scenario without building collapse (the black curve in Fig. 5h). The middle row is associated with the building collapse scenario with the largest inundation depth at station 15, whereas the bottom row represents the scenario with the smallest inundation depth. The red circle is the location of station 15.

(Fig. 5f), only 28% of the 100 scenarios show positive flood depth. With $r = 10\%$ and 20%, however, these numbers are 78% and 62% respectively (Table 2). A similar trend is observed at station 17, where a large $r$ reduces flood exposure (only 10 scenarios are flooded at $r = 40\%$).

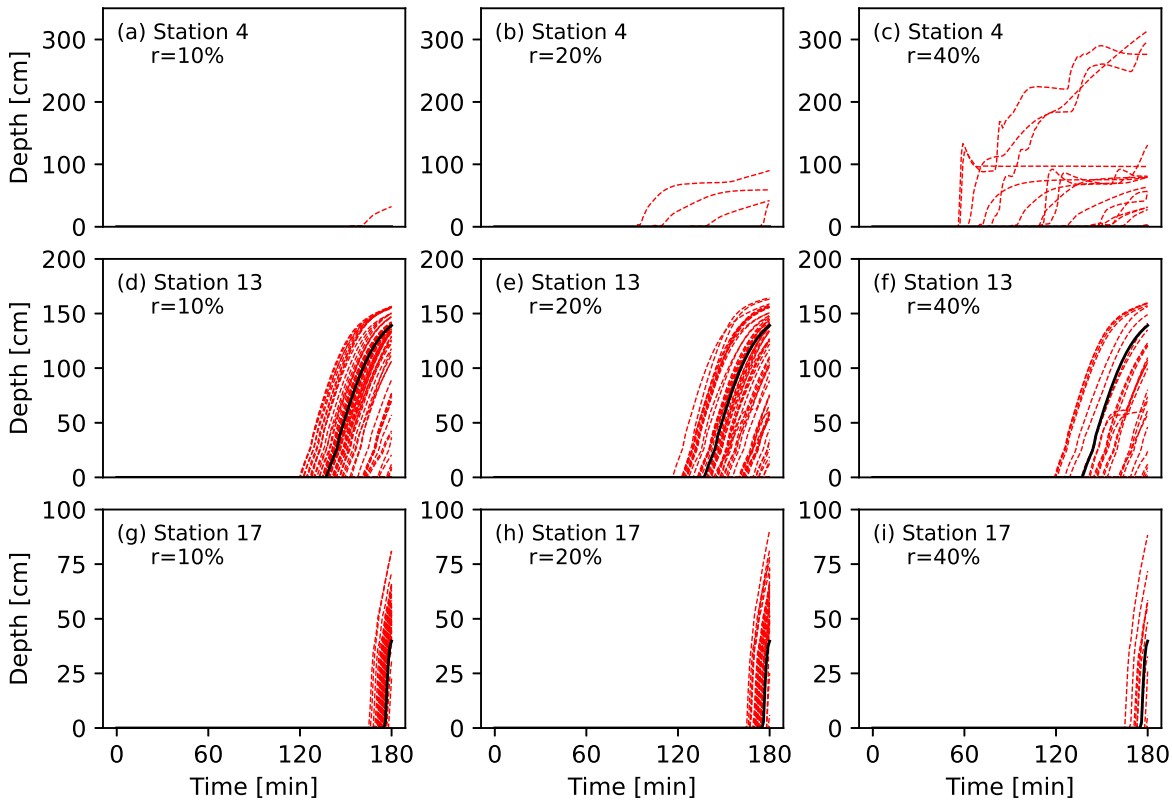

**Figure 7.** Water depth evolution at metro station 4, 13 and 17 for the fluvial flood scenarios. The black line is the original DEM without building collapse. The red lines are the 100 random realizations with buildings collapsed.

A notable difference between fluvial and pluvial flood is that, unlike pluvial flood scenarios where $p_{\text{flooded}}$ always reach

100%, for fluvial flood, both water depth and $p_{\text{flooded}}$ exhibit variability between different stations, different $r$ values, and different random realizations (Table 2). In terms of the mean deviation ($\epsilon_{\text{avg}}$), the trends of $\epsilon_{\text{avg}}$ and $p_{\text{flooded}}$ are the same for station 4 and 17. With more testing scenarios flooded, the mean deviation increases. However, this trend is reversed at station 13, where the lowest $p_{\text{flooded}}$ ($p_{\text{flooded}} = 28\%$ at $r = 40\%$) is accompanied with the largest deviation ($\epsilon_{\text{avg}} = 22.29$cm). Again, these results illustrate the complex interactions between flood propagation and the collapsed buildings.

Figure 8 shows three selected scenarios to better understand the flooding process of station 13. At the end of the simulation, the water depth at station 13 reaches about 140cm for the reference scenario. As can be seen from Fig. 8(b), the flood water originates from break point B (Fig. 2). The intruding water travel southward following the terrain slope. Then it changes direction towards the west when encountering elevated terrains, and turns back to south again when hitting buildings. After

**Table 2.** The mean deviation from the reference scenario, the proportion of scenarios being flooded, and the proportion of scenarios with aggravated flood disaster (i.e., greater water depth than the reference) among the 100 random building collapse and fluvial flood scenarios.

| | Station 4 | | | Station 13 | | | Station 17 | | |
|---|---|---|---|---|---|---|---|---|---|
| | $r=10\%$ | $r=20\%$ | $r=40\%$ | $r=10\%$ | $r=20\%$ | $r=40\%$ | $r=10\%$ | $r=20\%$ | $r=40\%$ |
| $\epsilon_{avg}$ [cm] | 2.30 | 4.32 | 10.02 | 13.72 | 20.02 | 22.29 | 2.11 | 1.60 | 1.36 |
| $p_{flooded}$ [%] | 1 | 4 | 17 | 78 | 62 | 28 | 61 | 33 | 10 |
| $p_{agg}$ [%] | | 100 | | 46 | 37 | 22 | 92 | 88 | 70 |

penetrating through a few building blocks, the river water finally reaches station 13. Figure 8(c) and (d) display the scenario with the deepest water depth at station 13. An obvious distinction is that some buildings along the Huangpu River are collapsed. The collapsed buildings squeeze the roads between them, which serve as a primary path of the intruding water. As a result, the flow velocity is higher and station 13 is flooded with greater depth. Figure 8(e) and (f) show the scenario with the lowest inundation depth at station 13. Although the flow path is squeezed by the collapsed buildings as well, additional buildings are collapsed next to station 13 in the northeast, which block the original southward flow path penetrating these buildings. As a result, the flooding of station 13 is weakened and postponed.

## 4    Discussions

As stated in Section 1, the purpose of this work is to understanding if, how and why building collapse affect the inundation of the metro stations. The massive simulation results clearly show that building collapse influence the inundation depth at metro station in the study area (Fig. 5 to 8). Furthermore, the inundation patterns vary between pluvial and fluvial floods, and vary from station to station. Thus, the following sections will discuss in more detail the mechanisms behind the observed variability in the flooding of metro stations, as well as the limitations of our approach.

### 4.1    Pluvial vs. Fluvial

From the results shown in Section 3, pluvial and fluvial floods exhibit distinct behaviors in the event of building collapses. Pluvial flood occurs in low-lying areas all around the study area. Building collapse do not alter the flood status ($p_{flooded}$ always equal 100%) of a given location (i.e., a metro station), but greater variance of the inundation depth is detected as more buildings collapse. Fluvial flood occurs close to the rivers. Its occurrence is sensitive to building collapse. As $r$ increases, the total number of flooded scenarios could either increase (station 4) or decrease (station 13 and 17).

The distinctions between pluvial and fluvial floods arise from their different origination, accumulation, and propagation patterns. Pluvial floods originate from heavy rainfall, which is spatially distributed. Rainwater accumulates in local topographic depressions, leading to massive short travel paths, which can be visualized from the velocity field of Fig. 6. Most flow paths remain unaffected by building collapse unless the collapsed building is in close proximity to the depression. As $r$ increases,

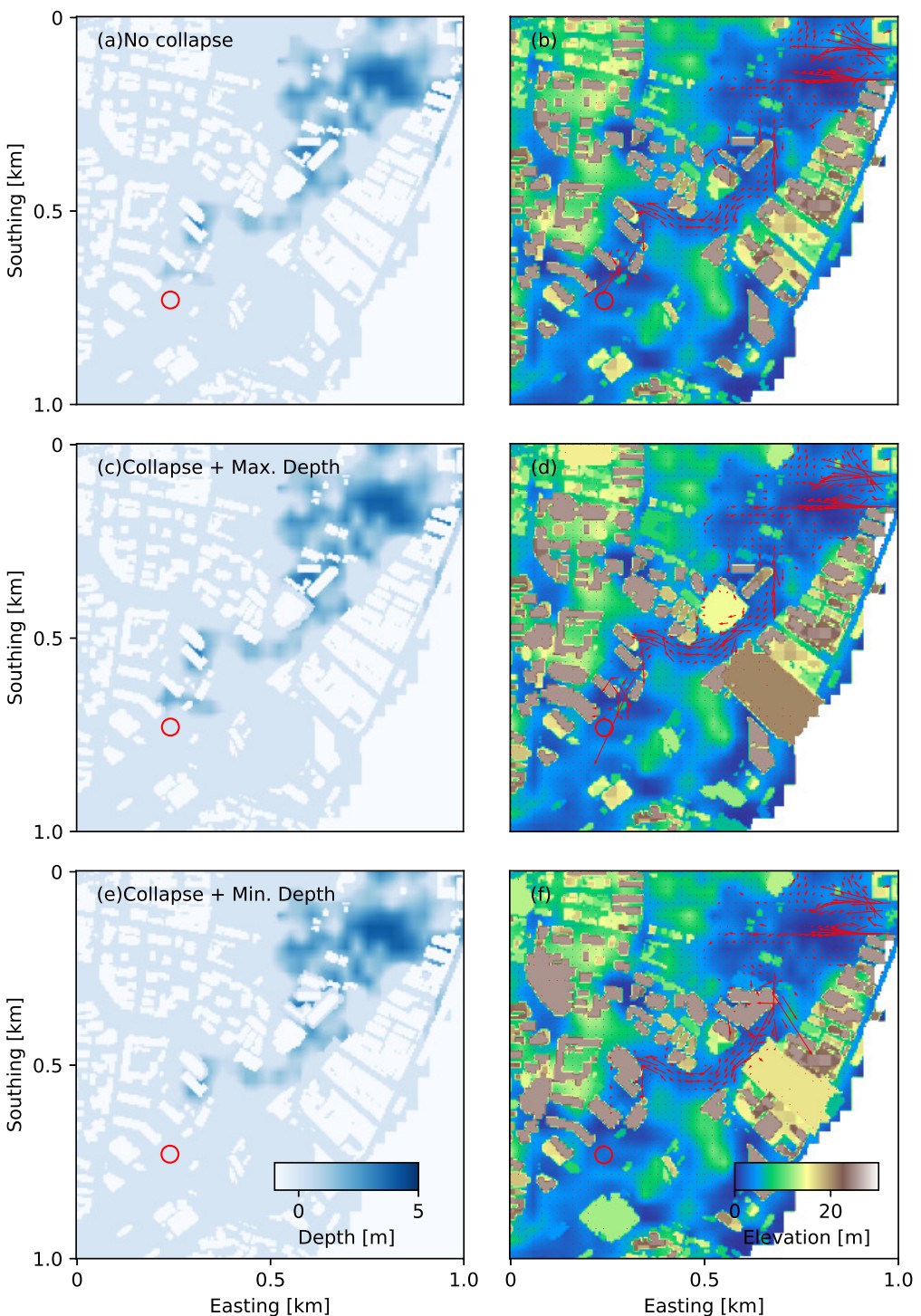

**Figure 8.** Simulated water depth (left column) and velocity (right column) at 2 hours for selected fluvial flood scenarios near metro station 13 (r=20%). The top row is the reference scenario without building collapse (the black curve in Fig. 7e). The middle row is associated with the building collapse scenario with the largest inundation depth at station 13, whereas the bottom row represents the scenario with the smallest inundation depth.

there is a greater probability that a chosen location is next to the collapsed building, which explains why more scenarios diverge from the reference at higher $r$.

On the contrary, fluvial flood originates from point sources (i.e., the break points along the river bank), forming only a few travel paths from the source. However, since the invading river water is driven by strong pressure gradient, it travels through longer distances. The long travel path implies a greater probability for the flow to be interfered by the collapsed building, and the very few paths means that once it is interfered, the subsequent flood propagation will be dramatically influenced. Thus, a large $r$ does not necessarily affect the inundation of a metro station, but when it does, the inundation states of the station could be completely altered.

## 4.2 Global vs. Local

Hydrodynamic simulation is the predominant method for urban flood studies as it replicates and forecasts the spatiotemporal evolution of inundation depth and flow fields. This capability allows for a detailed examination of the complete flooding process at finer spatial and temporal resolutions. However, this advantage is not always fully exploited as many urban flood studies focus on the "global" flood variables and the "global" influencing factors. The former are the variables that characterize flood of the entire study area (e.g., inundation extent, flood volume, mean water depth and outflow discharge), and the latter are the factors that are applied to the entire study area (e.g., rainfall intensity, building coverage, drainage capacity) (e.g. Bermúdez et al., 2018; Bruwier et al., 2020; David and Schmalz, 2020).

Buildings in megacities often exhibit strong spatial heterogeneity. Taking Shanghai as an example, historical buildings from the early 20th century, civil apartments from the post-World War II era, and some of the world's tallest skyscrapers coexist. These structures differ in age, height, construction, materials, usage, and earthquake resistance. Consequently, they are expected to react differently when subjected to seismic activity, indicating that earthquake-induced building collapse could be localized. Similarly, urban flood vulnerability is also localized. Metro stations are inherently more vulnerable than most other urban infrastructures. Thus, acknowledging that the collapse of a building at location A can hardly affect the flooding of location B that is kilometers away, this study adopts a local-to-local approach when analyzing results. For example, the water depth at each station is examined together with detailed building collapse patterns nearby (Fig. 6 and 8). If we only look at global variables, for example, the inundation extent shown in Fig. 8(a), (c) and (e), minimal differences are visible between different scenarios.

Herein, we want to emphasize the importance of performing high-resolution flood simulations with HPC-enabled hydrodynamic models, which allows investigating key localized flood processes. This is particularly important in terms of risk assessment and mitigation because it is often the local spots (metro stations, bridge tunnels, narrow channels, etc.) that suffer the most severe flood disasters (e.g. Hénonin et al., 2015; Vermeij, 2016).

### 4.3 Limitations

The present study is a preliminary attempt to explore how building collapses affect pluvial and fluvial flooding at metro stations. To achieve the research goal, we conducted 100 flood simulations with various random realizations of building collapse pat-

terns, allowing for a statistical examination of the consequences of building collapse on flooding, neutralizing any uncertainties in model parameters and simplified model treatments. Such uncertainties and simplifications arise from neglecting infiltration and drainage, using non-physical building collapse and levee breach models, assuming uniform rainfall intensity and omitting metro station structures. Herein, we illustrate that these simplifications do not affect our findings and analysis on how building collapse impacts flooding. The reported results remain significant for the development of more detailed, physics-based, local-to-local urban multi-hazard studies in the future.

The present study considers surface hydrodynamic processes of pluvial and fluvial flood events. Urban flood also involves infiltration into the subsurface, and drainage through the stormwater pipes (e.g. Bermúdez et al., 2018; Cardoso et al., 2020; Hossain Anni et al., 2020; Schubert et al., 2022b). These two processes are neglected in this work due to the lack of data on the geological conditions, the water table and the pipe layout of the study area. Indeed, data scarcity is a major challenge when incorporating underground infrastructures with surface hydrodynamic simulations. Although empirical methods are available to approximate the amount of soil infiltration and pipe drainage (e.g. Bruwier et al., 2020; Xu et al., 2023), they are not applied herein because Shanghai as a coastal city features very shallow water table (in the range of 1-2m). Under extreme flooding conditions the soil could reach saturation rapidly. The pipes could reach their full capacity too. Figure 5 demonstrates that the maximum inundation depth of the July 21th flood ranges between 30 to 60cm among the metro stations, which is higher than the reported values in the media (around 25cm), but within a reasonable range considering the absence of drainage pipes. We argue that the mechanism how building collapse affect flood propagation, which is the focus of this work, is not sensitive to soil infiltration or pipe drainage. As part of the ongoing work, SERGHEI-SWE is being developed to integrate a variably-saturated groundwater solver, and a stromwater drainage model. The fully coupled surface-subsurface-pipe flow model should be able to provided a physically more complete picture of the urban flooding process in the future.

In this work, building collapse is simplified as changes in the building geometry, and earthquake vulnerability (i.e., which building will collapse during an earthquake) is assigned randomly. This method treats earthquake and building strength as "global" variables, which is not aligned with our local-to-local strategy (Section 4.2). Indeed, more advanced approaches inevitably require detailed information on the buildings in the study area, including the building height, age, usage, material and so on (Xin et al., 2021). We argue that such information are not necessary for the present study because we put more attention on the flood propagation mechanisms and processes. However, in future multi-hazard studies, these data should be obtained and integrated with advanced building collapse models to provide more realistic, city-wide building collapse estimations. Similar to building collapse, modeling of the levee breach process is also omitted. We randomly selected three breach points that are sufficiently spaced apart to allow floodwaters to reach the interior of the study area as much as possible, and that the floodwaters from the breach points do not interact with one another. Future studies should consider the strength and vulnerability of the flood wall in greater detail.

In the present study, metro stations are simplified to single pixels on the DEM. However, real metro stations in Shanghai often contain multiple exits spanning several road blocks. Consequently, the flooding status and flood resistance at each exit could vary. Although further refining the grid resolution to resolve the local topography at each metro exit remains challenging, a multi-scale approach might be feasible in the future. For example, based on the hydrodynamic simulation results at relatively

coarse grid resolutions (e.g., the 5m resolution used herein), a finer resolution simulation could be performed in a smaller region near the metro station. This would characterize the detailed flooding processes at each exit. The multi-scale modeling approach, coupled with a physics-based building collapse model, would enable a true local-to-local analysis and evaluation of the multi-hazard risk posed to metro stations.

In this work, rainfall is assume to be spatially uniform. Although city-scale spatial heterogeneity of rainfall intensity and duration has been reported (e.g., Zhuang et al., 2020), it is not considered in this study because (i) incorporating rainfall heterogeneity would significantly increase the amount of simulation scenarios required, and (ii) the study area is relatively small (about 20 km$^2$). Furthermore, by adding rainfall heterogeneity as an additional variable, the focus of this manuscript would shift from examining the relationship between building collapse and flooding, to exploring the relationship between rainfall heterogeneity and flooding. Thus, we choose to use the measured rainfall data from a single rainfall event, and ignore the spatial distribution of the rainfall.

Finally, we want to emphasize that the present work focuses exclusively on the hazard aspects of urban flooding, specifically examining the physical mechanisms by which building collapse affects urban flooding. An evaluation of risk, vulnerability, and urban resilience is beyond the scope of this study. Future research should incorporate both hazard and risk analysis to provide robust guidance for enhancing urban resilience against multi-hazard events. In particular, in alignment with the previously discussed local-to-local strategy, a more detailed investigation into the influence of local vulnerabilities in key infrastructures is required.

## 5    Conclusions

In this study, GPU-accelerated hydrodynamic simulation is employed to simulate pluvial and fluvial flood propagation before and after earthquake-induced building collapse, aiming at understanding the mechanism how building collapse influences flood inundation of urban metro stations. Taking Huangpu district in central Shanghai as the study area, the following conclusion are drawn from the simulation results:

1. Pluvial flood

   Pluvial flood occurs over a broad spatial extent. The travel paths of the flood water are relatively short due to the widely distributed local topographic depressions that serve as natural drainage points. Thus, only the buildings collapsed near the metro station can affect its inundation by interfering the relevant propagation paths of the water that floods the station. It also follows that the impact of building collapse on metro station flooding is positively correlated to the proportion of buildings that have collapsed.

2. Fluvial flood

   Fluvial floods originate from point sources such as a breaching levee. The propagation of fluvial flood is characterized by long and consistent trajectories spreading from the source and following the topographic gradient. Building collapse only affects fluvial flood when the collapsed buildings are located close to the propagation path of the flood water. However,

since these paths are generally longer than those for pluvial floods, the inundation of a metro station could be influenced by buildings collapsed some distance away. Moreover, an increasing proportion of building collapse does not necessarily enhance the flood risk because there is a large probability that the collapsed buildings are situated far from the flood source or its travel path. The key factor is the spatial relationship between the flood propagation path, the location of building failures and the location of the hazard-bearing body of interest (e.g., a metro station).

This study provides novel insights into the complex compound mechanism involving earthquake-induced building collapse and subsequent flooding under a multi-hazard context. While the findings contribute to advancing multi-hazard evolution process and risk analysis, it is important to acknowledge that the explicit modeling of earthquake processes and their physical impacts on individual buildings was beyond the scope of this work. Future research efforts should focus on developing comprehensive physical-process-based multi-hazard simulators capable of capturing the intricate dynamics of such compound events at a finer scale.

*Code and data availability.* TEXT

The SERGHEI-SWE model is an open-sourced software (Caviedes-Voullième et al., 2023). The topography, building and road layout data of the study area are all publicly available. The rainfall data is measured by the authors and is available upon request.

*Author contributions.* TEXT

ZL conceptualized the research aim and scope; ZZ prepared the data; ZL performed the numerical simulation; ZL and HL analyzed the results; ZL and HL wrote the manuscript draft; CD and SJ reviewed and edited the manuscript.

*Competing interests.* TEXT

The authors declare no competing interests.

*Acknowledgements.* This work has been supported by the National Key R&D Program of China (2022YFC3803000), the Fundamental Research Funds for Central Universities (China).

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
