# Peer review of "Influence of building collapse on pluvial and fluvial flood inundation of metro stations in central Shanghai"

_EGUsphere, 2024_

## Referee Comment (RC2)

[referee-annotated manuscript omitted]

---

## Author Comment (AC1)

**Response to Referee 1 Comments on:**

**Influence of building collapse on pluvial and fluvial flood inundation of metro stations in central Shanghai**

Zhi Li, Hanqi Li, Zhibo Zhang, Chaomeng Dai, and Simin Jiang

Thank you for your thorough review of our manuscript. We appreciate your constructive feedback and we have addressed your comments point by point in our revision.

**Referee #1 (RC1):**

**General Comments**

1. However, the discussion should further clarify the novelty of this work, particularly in relation to the specific objectives defined in the introduction. I suggest revising the initial section to define the aims and objectives more explicitly and structuring the discussion to reflect these objectives (authors state "We aim at understanding if, how and why earthquake-induced building collapse affect the inundation of the metro stations. The findings will serve as the cornerstone for multi-hazard risk assessment of urban metro systems").

**Response:** We agree that the objectives need to be more explicitly stated and the novelty of our work should be further clarified. We have revised the Introduction and the Discussions sections to explicitly state our aims and objectives:

*Introduction: "Prevailing urban flood studies focus on the global inundation depth and extent. However, collapsed buildings and flooded metro stations are localized. It is rarely investigated that how do localized building collapse affect the global flood propagation, and then subsequently affect local inundation of metro stations? In this study, we aim at understanding if, how and why building collapse affect the inundation of the metro stations, with a focus on the spatial heterogeneity of both the building collapse and the station water level, thus addressing the local-to-local multi-hazard process."*

*Discussions: "the purpose of this work is to understanding if, how and why building collapse affect the inundation of the metro stations. The massive simulation results clearly show that building collapse influence the inundation depth at metro stations in the study area (Fig. 5 to Fig. 8). Furthermore, the inundation patterns vary between pluvial and fluvial floods, and vary from station to station. Thus, the following sections will discuss in more detail the mechanisms behind the observed variability in the flooding of metro stations."*

2. It is important to specify that the analysis focuses on hazard rather than risk, as no vulnerability assessments of the exposed assets were considered. The manuscript correctly discusses multi-hazard rather than multi-risk; however, a more detailed description emphasizing this

distinction is recommended. Additionally, it is crucial to note in the discussion that the call for more local-scale analysis, highlighted in the article, must be accompanied by a detailed analysis of both exposure and vulnerability. Otherwise, the effort in hazard assessment may be rendered ineffective. Moreover, greater attention should be given to the term multi-hazard: the work is preparatory for multi-hazard assessment, as the earthquake is not studied directly in this work but rather random collapses are considered.

**Response:** Thank you for highlighting the importance of distinguishing between hazard and risk. We have revised the Introduction and the Discussions sections to clearly state that our analysis focuses on hazard rather than risk. We have further clarified that while the work lays the groundwork for understanding multi-hazard scenarios, it does not encompass a full risk assessment.

*Introduction: "However, it should be noted that (i) our analysis focuses on the hazard rather than the risk aspect of building collapse and urban flood. An in-depth exploration of improved risk assessment techniques is beyond the scope of the present manuscript, and (ii) building collapse is used to represent the consequence of earthquake, but the physical processes of seismic events are not explicitly modeled. With this aim, topics such as how the earthquake is triggered, how the buildings are collapsed, how the flood enters the metro station and how to assess the multi-hazard probability and risk are beyond the scope of this manuscript."*

*Discussions: "Finally, we want to emphasize that the present work focuses exclusively on the hazard aspects of urban flooding, specifically examining the physical mechanisms by which building collapse affects urban flooding. An evaluation of risk, vulnerability, and urban resilience is beyond the scope of this study. Future research should incorporate both hazard and risk analysis to provide robust guidance for enhancing urban resilience against multi-hazard events. In particular, in alignment with the previously discussed local-to-local strategy, a more detailed investigation into the influence of local vulnerabilities in key infrastructures is required."*

**Specific Comments**

1. Figure 1 appears incomplete and does not describe the entire complexity. If it is self-produced, the various choices need to be explained and justified; otherwise, the source should be cited (e.g., for river the driver is only SLR, bearing body are only those?). I do not believe it helps clarify the manuscript's work; I suggest rethinking it to focus more on the conducted work rather than a general framework.

**Response:** After careful consideration, we decide to remove Figure 1 because as the reviewer said, Fig. 1 does not help to clarify the work, and it could be misleading. We believe the purpose, scope and procedures of this work have been clearly illustrated using texts.

2. Add a discussion on the assumption of constant rainfall over the entire area of study: while it is a cautious approach, is it reasonable? This should be discussed in the manuscript.
Line 236: "Pluvial floods originate from heavy rainfall, which is spatially distributed" -> This is due to the assumption of constant rainfall, but in reality, rainfall varies from area to area and

maybe the difference from the fluvial flood will become less relevant.

**Response:** We have added a paragraph in the Discussions section to illustrate why we assume uniform rainfall intensity:

*"In this work, rainfall is assume to be spatially uniform. Although city-scale spatial heterogeneity of rainfall intensity and during has been reported, it is not considered in this study because (i) incorporating rainfall heterogeneity would significantly increase the amount of simulation scenarios required, and (ii) the study area is relatively small (about 20 km$^2$). Furthermore, by adding rainfall heterogeneity as an additional variable, the focus of this manuscript would shift from examining the relationship between building collapse and flooding, to exploring the relationship between rainfall heterogeneity and flooding. Thus, we choose to use the measured rainfall data from a single rainfall event, and ignore the spatial distribution of the rainfall."*

3. Line 127: What magnitude scale was used to indicate the earthquake?

**Response:** The paper of Cole et al. 2008 used surface wave magnitude and Chinese Intensity Scale. We have added this information in the manuscript:

*"Historically, Shanghai rarely experiences an earthquake with a surface-wave magnitude greater than 7.0."*

*"It is estimated that in the Nanjing Road (located in Huangpu district) where plenty of old houses are made of brick and timber, 7.2% of buildings will be seriously damaged under an intensity VI earthquake (Chinese Intensity Scale), and 24.8% of buildings will be seriously damaged under an intensity VII earthquake."*

4. Line 140: Provide more information on the collapse mechanism. Although the model is cited, it would be helpful to provide a few more details on the adopted collapse mechanism. For instance, how is the presence of nearby buildings considered? How far do the debris reach?

**Response:** We have provided more detailed description of the building collapse mechanisms and the estimation method considered in our analysis, including the treatment of nearby buildings. We have added a new figure to assist the illustration:

*"building collapse is modeled by distributing certain amount of the ``building material" to its surroundings, thereby increasing the spatial occupancy and reducing the height of the building. In this study, we assume that when a building collapses, (i) its spatial occupancy doubles, and (ii) its total volume remains unchanged. The increased spatial occupancy (i.e., the ``debris") is distributed uniformly around the building. The reduced height of the building can be estimated using the volume and the new spatial occupancy. A special case exists when two buildings are adjacent and their spatial occupancy overlap when collapsed. In such situations, the two buildings are treated as one single building when calculating the debris extent. It should be noted that if two buildings with different heights are treated as one building, the final building height after collapse could be greater than its original height, which is unrealistic (Fig.3). However, this phenomenon has negligible influence on the subsequent flood simulation because as long as the building debris is not inundated, slight variation of its height has minor influence on the flow field. That is, the*

*obstruction of the flow path (due to the spreading of the debris extent) has much stronger influence on flood propagation than minor changes of the debris height. With this approach, building collapse is only a simplified representation of the consequence of hypothetical seismic events, but the physical mechanism connecting earthquake and building collapse is not involved."*

[Figure]

5. Line 170: The collapse mechanism becomes crucial for considering the impact on a station located within a collapsing building.

**Response:** We agree that when a station is located within a collapsed building, the collapse mechanism becomes important. However, we think this importance is more from the perspective of vulnerability and risk analysis. Our study mainly focuses on the flood propagation process and mechanism, that is, the hazard itself. Thus, despite that we pointed out the issue of station-in-a-building, more detailed analysis on this issue is beyond the scope of this manuscript. We have added a few sentences to explain our reasons:

*"It implies that for a complete analysis of multi-hazard risk of the metro stations, the locations of each entrance should also be considered separately because the location affects station vulnerability under earthquake-induced building collapse, but vulnerability analysis is beyond the scope of the present study."*

6. Line 213: The written content does not correspond to what is reported in Table 2; please verify and correct.

Response: We appreciate the reviewer for pointing out the inconsistency between Table 2 and its description. Indeed, we described the number of flooded scenarios in the paragraph, but reported the percentage of flooded scenarios in the Table. We have unified the paragraph and the Table to

both use percentages.

Thank you again for your constructive comments.

Sincerely,
Zhi Li (Corresponding Author)

---

## Author Comment (AC2)

**Response to Referee 2 Comments on:**

**Influence of building collapse on pluvial and fluvial flood inundation of metro stations in central Shanghai**

Zhi Li, Hanqi Li, Zhibo Zhang, Chaomeng Dai, and Simin Jiang

Thank you for your constructive feedback on our article regarding the impact of building collapse on flooding of metro stations. We appreciate your insights and agree that several areas need clarification and expansion. Below, we outline how we plan to address each point raised in your review.

**Referee #2 (RC2):**

1. When I finished to read the introduction (chap. 1) it did not become clear how the building collapse is taken into account; this should at least be addressed briefly here and then refer to chap 2.3 for more details; add reference(s) to the literature.

**Response:** We have revised the Introduction section to include a brief discussion on how building collapse is represented into our model:

*"However, we do not directly model the building collapse processes. Instead, building collapse is represented as the spreading of the debris that changes the building height and spatial occupancy. The total volume of a building remains unchanged before and after it collapses. For a more detailed description of how building collapse is treated, please refer to Section 2.3 and Takabatake et al. (2022)."*

2. Chap 2.2: how are the metro stations taken into account, as holes in the model, i.e. inner open boundaries, where water can flow out of the domain? If I see correctly, this is not the case and you just observe the water level in the cells where metro stations are; this simplification should be discussed and explained, possibly you already comment on this in chap. 1.

**Response:** Indeed, we do not model water flow into the metro stations. We only observe the water level in the cells where the station is located. We have added a brief discussion in Section 2.2 on the pros and cons of this approach. We have also discussed in Section 4.3 on the possible improvements of the treatment of metro stations in future works.

*Section 2.2: "It should be noted that a real metro station often has multiple exits, each with different elevation and orientation that cannot be adequately resolved at the grid resolution we use. Thus, in our study, we do not treat metro stations as sink terms or outflow boundaries. We only observe the water depth of the grid cell where the metro station is located. The possibility of further refining the treatment of metro stations will be discussed in Section 4.3."*

*Section 4.3: "In the present study, metro stations are simplified to single pixels on the DEM. However, real metro stations in Shanghai often contain multiple exits spanning several road*

*blocks. Consequently, the flooding status and flood resistance at each exit could vary. Although further refining the grid resolution to resolve the local topography at each metro exit remains challenging, a multi-scale approach might be feasible in the future. For example, based on the hydrodynamic simulation results at relatively coarse grid resolutions (e.g., the 5m resolution used herein), a finer resolution simulation could be performed in a smaller region near the metro station. This would characterize the detailed flooding processes at each exit. The multi-scale modeling approach, coupled with a physics-based building collapse model, would enable a true local-to-local analysis and evaluation of the multi-hazard risk posed to metro stations."*

3. 141: The sentencse does not become clear, more explanation is required.

   chap. 2.3: what happens during collapse is not fully clear; better description or a figure where 1 building is shown before and after collapse, give length, width, height, volume etc.

**Response:** The third and fourth comments are both related to the description of the building collapse mechanism. We have revised Section 2.3 and added a figure to address the reviwers' concerns :

*"building collapse is modeled by distributing certain amount of the ``building material'' to its surroundings, thereby increasing the spatial occupancy and reducing the height of the building. In this study, we assume that when a building collapses, (i) its spatial occupancy doubles, and (ii) its total volume remains unchanged. The increased spatial occupancy (i.e., the ``debris'') is distributed uniformly around the building. The reduced height of the building can be estimated using the volume and the new spatial occupancy. A special case exists when two buildings are adjacent and their spatial occupancy overlap when collapsed. In such situations, the two buildings are treated as one single building when calculating the debris extent. It should be noted that if two buildings with different heights are treated as one building, the final building height after collapse could be greater than its original height, which is unrealistic (Fig.3). However, this phenomenon has negligible influence on the subsequent flood simulation because as long as the building debris is not inundated, slight variation of its height has minor influence on the flow field. That is, the obstruction of the flow path (due to the spreading of the debris extent) has much stronger influence on flood propagation than minor changes of the debris height. With this approach, building collapse is only a simplified representation of the consequence of hypothetical seismic events, but the physical mechanism connecting earthquake and building collapse is not involved."*

[Figure]

4. You should more clearly state that your approach is a very first step to investigate impacts of building collapse on flooding at metro stations; what is the value of your results so far; you arbitrarily selected n=100 simulations with different number of collapses, what happens, if the number n is higher; you arbitrarily selected 3 break points for the fluvial flooding, what happens if this number differs; there is no need to make further simulations, but you should discuss this (more), include that in chap. 4, overall more justification.

**Response:** We have incorporated a more robust discussion in Section 4.3 regarding the implications of our simplifications and assumptions. We believe the scope and the limitations of our work is more clear now.

*"The present study is a preliminary attempt to explore how building collapses affect pluvial and fluvial flooding at metro stations. To achieve the research goal, we conducted 100 flood simulations with various random realizations of building collapse patterns, allowing for a statistical examination of the consequences of building collapse on flooding, neutralizing any uncertainties in model parameters and simplified model treatments. Such uncertainties and simplifications arise from neglecting infiltration and drainage, using non-physical building collapse and levee breach models, assuming uniform rainfall intensity and omitting metro station structures. Herein, we illustrate that these simplifications do not affect our findings and analysis on how building collapse impacts flooding. The reported results remain significant for the development of more detailed, physics-based, local-to-local urban multi-hazard studies in the future."*

*"Similar to building collapse, modeling of the levee breach process is also omitted. We randomly*

*selected three breach points that are sufficiently spaced apart to allow floodwaters to reach the interior of the study area as much as possible, and that the floodwaters from the breach points do not interact with one another. Future studies should consider the strength and vulnerability of the flood wall in greater detail."*

*"Finally, we want to emphasize that the present work focuses exclusively on the hazard aspects of urban flooding, specifically examining the physical mechanisms by which building collapse affects urban flooding. An evaluation of risk, vulnerability, and urban resilience is beyond the scope of this study. Future research should incorporate both hazard and risk analysis to provide robust guidance for enhancing urban resilience against multi-hazard events. In particular, in alignment with the previously discussed local-to-local strategy, a more detailed investigation into the influence of local vulnerabilities in key infrastructures is required."*

5.some more information on OpenMP, Cuda, HIP, MPI; not everyone knows this.
**Response:** Thanks for the advice. We have added explanations and details of the above professional designations in Section 2.2:

*"Without modifying the source codes, SERGHEI-SWE can perform parallel computation on a CPU through OpenMP, or on a Graphical Processing Units (GPUs) through either CUDA (a parallel computing model for Nvidia GPUs) or HIP (a C++ kernel language for parallel computing on Nvidia and AMD GPUs). This feature is achieved through the Kokkos framework. It also supports distributed memory parallelization through the Message Passing Interface (MPI), which allows parallel computation across multiple CPU or GPU nodes."*

6.in Tab 1, 2: give [%] here, then no need to repeat in each row
**Response:** We have modified Table 1 and 2 to put [%] in the first column.

Further minor comments are in the attached pdf.
**Response:** We have carefully fixed the typos and the grammar mistakes the reviewer pointed out in the PDF. They are not listed one-by-one here but they should be reflected in the revised manuscript.

We appreciate your insightful suggestions and believe these revisions will significantly enhance the clarity and depth of our paper. Thank you once again for your thorough review.

Sincerely,
Zhi Li (Corresponding Author)